# Real World Analysis of Quality of Life and Toxicity in Cancer Patients Treated with Hyperthermia Combined with Radio(chemo)therapy

**DOI:** 10.3390/cancers15041241

**Published:** 2023-02-15

**Authors:** Adela Ademaj, Emsad Puric, Olaf Timm, David Kurti, Dietmar Marder, Thomas Kern, Roger A. Hälg, Susanne Rogers, Oliver Riesterer

**Affiliations:** 1Center for Radiation Oncology KSA-KSB, Cantonal Hospital Aarau, 5001 Aarau, Switzerland; 2Doctoral Clinical Science Program, Medical Faculty, University of Zürich, 8032 Zürich, Switzerland; 3Institute of Physics, Science Faculty, University of Zürich, 8057 Zürich, Switzerland

**Keywords:** superficial hyperthermia, deep regional hyperthermia, radiochemotherapy, quality of life, toxicity symptoms, curative intent, palliative intent

## Abstract

**Simple Summary:**

Hyperthermia (HT) is a clinical treatment modality that is used in combination with radiotherapy (RT) and/or chemotherapy (CT) in cancer patients to enhance the effect of ionizing radiation and chemotherapeutic drugs without worsening toxicity. Yet there is very little clinical evidence about the quality of life (QoL) after combination treatments, including HT, despite the fact that QoL has gained increasing importance in assessing the side effects of oncological therapies. In this retrospective study, the QoL of patients treated with HT in combination with radio(chemo)therapy (RCT) with curative or palliative intent was assessed by the EORTC QLQ-C30 questionnaire. The results of this analysis suggested that the combination of RCT and HT stabilizes or improves QoL for a clinically relevant time period after treatment.

**Abstract:**

Hyperthermia (HT) in combination with radio(chemo)therapy (RCT) is a well-established cancer treatment strategy. This report analyses the quality of life (QoL), toxicity and survival outcomes in patients with different tumor entities who received HT in combination with RCT. The primary endpoint of this study was the assessment of QoL scale items 3 and 12 months after treatment in patients who were treated with palliative intent and curative intent, respectively. The secondary endpoints of this study were acute toxicities, 1-year overall survival (OS), and local progression-free survival (LPFS). Patients treated with curative intent experienced significant improvement in emotional functioning (EF), social functioning (SF), financial difficulties (FI) and insomnia (SL) 12 months after treatment. Patients had significantly improved FI and pain (PA) three months after palliative treatment. Acute toxicity of grade 3 or more was 26% during treatment and 4% after three months. The 1-year OS rates were 90% (95% CI: 79–96%) and 44% (95% CI: 31–59%) for patients treated with curative and palliative RCT combined with HT, respectively. Moreover, the 1-year LPFS rates were 94% (95% CI: 84–98%) for patients treated with curative intent and 64% (95% CI: 50–77%) for palliative patients. In summary, combined RCT and HT stabilized or improved QoL scale items for both curative and palliative indications.

## 1. Introduction

Hyperthermia (HT) is a cancer treatment modality that aims to heat cancer cells to 40–43 °C, thereby inducing cytotoxic effects and sensitizing the cells for radiotherapy (RT) and chemotherapy (CT) [1,2]. Several randomized clinical studies showed that the addition of HT to RT and/or CT improves clinical outcomes for different cancer entities treated with palliative or curative treatment intention without inducing significant side effects [3,4,5,6,7].

Depending on the cancer entity and the treatment site, HT treatment is delivered using a superficial or a deep regional HT device. The specifications, characteristics, performance and limitations of superficial and deep regional HT techniques have been summarized in detail elsewhere [8,9,10]. During the cancer treatment course, HT is delivered before or after RT once or twice per week, separated by at least 72 h due to the thermotolerance effect, which renders cancers insensitive to heat [11]. High temperatures achieved during HT sessions are reported to be associated with improved clinical outcomes [12,13,14]. However, temperature was reported not to correlate with an increase in acute side effects and skin toxicity in patients treated with superficial HT [15]. A retrospective study revealed no significant difference in the incidence of acute or late grade III toxicity between patients who achieved low or high temperatures with HT [16,17]. The clinical analysis showed that the only symptomatic toxicity that tended to associate with HT was pain during treatment [13]. Overall, no significant acute or late grade III toxicities were reported to be induced by HT [18,19,20,21].

Health-related quality of life (QoL) is an important prognostic factor in patients with cancer [22,23]. Although there is substantial clinical evidence that HT improves clinical outcomes when combined with RT or CT, there is still little knowledge about the effects of HT on the QoL of cancer patients when integrated into curative and palliative treatment concepts. Some of the few available QoL data analyses were reported from prospective phase II studies in advanced rectal cancer, bladder cancer and biochemically recurrent prostate cancer treated with deep HT [24,25,26]. These analyses show good QoL; however, the results should be interpreted with caution because the analyses were either reported in very small sample sizes or in cancer-free patients only.

This study aims to extend our knowledge in cancer patients treated with curative or palliative radio(chemo)therapy (RCT) combined with HT by evaluating QoL using the European Organization for Research and Treatment of Cancer (EORTC) QLQ-C30 questionnaire at different time points after treatment under real-world conditions. The term “real-world conditions” refers to a patient cohort where all patients treated with hyperthermia in our center were prospectively included, and data were routinely collected from a variety of sources. The benefits and limitations of real-world analyses in comparison to randomized clinical trials have been summarized elsewhere [27]. To complement the findings, acute toxicity symptoms and 1-year OS and LPFS were also analyzed.

## 2. Materials and Methods

### 2.1. Study Design and Population

This was a retrospective, single-center analysis of cancer patients treated with RCT combined with superficial HT or deep regional HT in the Radiation Oncology Center at Cantonal Hospital Aarau between April 2018 and October 2021. The inclusion criteria for this analysis were patients who received superficial or deep regional HT (≥2 HT sessions) combined with external photon beam RT and who returned the QoL questionnaires before treatment, immediately after treatment, at 3 and 12 months after treatment.

### 2.2. Hyperthermia Treatment

Superficial and deep regional HT were performed according to European Society for Hyperthermic Oncology (ESHO) guidelines [9,28]. For all patients treated with superficial and deep regional HT, the temperature metrics, such as average temperature achieved in measurement points (Tavg) and maximum temperature achieved at measurement points (Tmax), were assessed. In addition, thermal dose CEM43 was calculated, including the time interval between the start of HT sessions and RT treatment.

#### 2.2.1. Superficial Hyperthermia Treatment

Superficial HT was administered using a BSD 500 applicator (BSD Medical Cooperation/Pyrexar, Salt Lake City, UT, USA) at a frequency of 915 MHz. Depending on the size of the superficial cancer, a surface applicator with 3, 8 or 24 spiral antennas covering diameters of 9 cm, 12 cm or an area of 24× 20 cm^2^ was selected. The temperature probes were placed between the skin and the water bolus (filled with deionized water) to monitor the temperature on the surface of the target. After a 15–30 min preheating phase, a therapeutic temperature between 40 and 43 °C was applied for one hour. Patients received superficial HT sessions once or twice per week with a median time interval of 14 (range: 6–87) min before RT.

#### 2.2.2. Deep Regional Hyperthermia Treatment

For the treatment of deeply seated cancers, a deep regional HT technique was applied using the BSD 2000/3D system combined with the Sigma-60 or Sigma-Eye phased array applicator (BSD Medical Cooperation/Pyrexar, Salt Lake City, UT, USA) at the frequency range of 75–120 MHz. The Sigma-60 and Sigma-Eye are annular phased array applicators that are comprised of a clear plastic shell, eight radiating dipoles, and a bolus membrane. The difference between Sigma-60 and Sigma-Eye applicators is the shape of the applicators’ plastic shell, as Sigma-60 and Sigma-Eye applicators have cylindrical and elliptical shapes, respectively. The choice of applicators was based on the patient’s size. Typically, the Sigma-Eye was used for smaller-sized patients.

For thermometry, temperature probes were placed in the rectum or bladder or vagina (for female patients) and on the anal margin. The temperature was measured at 10-s intervals, starting before treatment and stopping immediately or five minutes after switching off the radiofrequency power. Thermal mapping was performed every 5 min with a step size of 1 cm and a maximum map length of 18 cm. After a 15–30 min preheating phase, a therapeutic temperature between 40 and 43 °C was applied for one hour. Patients were treated with deep regional HT sessions once or twice per week with a median time interval of 16 (range: 11–83) min before RT.

### 2.3. Electronic Patient Records

The hospital information system KISIM (CISTEC) and oncology information system ARIA (Varian Medical Systems) were used to extract general patient data (age, sex, Karnofsky performance index (KPI) before treatment), disease characteristics (type and TNM staging of primary tumors, treatment site, and recurrence disease status). Additionally, we extracted detailed treatment specifications such as RT treatment parameters (RT total dose, RT dose per fraction, and boost RT dose (if delivered)), HT treatment parameters (total number of HT sessions, time interval between HT and RT, temperature metrics, and CEM43). It was also documented for all patients whether they received CT in combination with HT and RT. The toxicity symptoms and clinical outcomes were evaluated based on medical follow-up reports 3 months and 12 months after treatment.

### 2.4. QoL Assessment and Secondary Outcomes

QoL was assessed by the EORTC QLQ-C30 questionnaire [29]. The QLQ-C30 comprises one Global Health Status scale, five functioning scales and nine symptom scales, as shown in Figure 1.

The scoring procedures were conducted as previously described by averaging items within three scales and transforming mean scores linearly [30]. A high score on the Global Health Status and functional scales represents better function. In the symptom scales, however, a higher score reflects a higher level of symptom burden. All patients were asked to complete the EORTC QLQ-C30 before treatment, after treatment (last day of HT), and after 3 and 12 months of follow-up. The outcome measure of this study was any change in QoL before and 12 months after curative treatment and any change in QoL before and 3 months after palliative treatment. Additionally, these outcomes at 12 months after treatment with curative intent and at 3 months after treatment with palliative intent were compared with the reference population of the EORTC QLQ-30 cohort [31].

The acute toxicity symptoms were also collected from medical reports for each individual patient during treatment and three months after treatment. The toxicity symptoms were evaluated using the Common Terminology Criteria for Adverse Events (CTCAE) version 4.0. Additional secondary outcomes were overall survival (OS) and local progression-free survival (LPFS). LPFS was calculated between the start of the last fraction of RT and the first of the following events: locoregional or metastatic recurrence from any cause, whereas OS was defined as the time period between the last fraction of RT and the date of death from any cause. Events were censored if the information on time to event was not available due to loss to follow-up or if the outcome event did not occur before one year.

### 2.5. Statistical Analysis

All clinical and treatment data were recorded in Microsoft Excel (version 16.0). Descriptive statistics were calculated for all variables under study. Statistical differences between continuous factors for different groups of patients were assessed using the Student’s *t*-test and Mann–Whitney *U* test after testing the normal distribution of the data. The primary outcome of the QoL analysis was the Global Health Status score, and the secondary outcomes were functional and symptom scale items. The missing data were excluded from the analysis, and the comparison of EORTC QLQ-C30 items scores between two different time points was performed in the same group of patients. The survival outcomes were analyzed using the Kaplan–Meier method. A *p*-value of ≤0.05 was considered statistically significant. All statistical analyses were conducted using R programming (version 4.1) with the following main libraries: tidyr, ggplot2, tidyverse, ggthemes, survival, survminer, and dplyr.

## 3. Results

### 3.1. Patient and Treatment Characteristics

A total of 97 patients with different tumor entities and treated either with curative (*n* = 45) or palliative (*n* = 52) RCT and HT in a single institution from April 2018 to October 2021 were included in this retrospective analysis. With regard to the treatment modality, 51% (49/97) and 49% (48/97) of patients were treated with superficial or deep HT, respectively. A small group of patients (15%) also received CT in combination with RT and deep HT. Baseline patient, disease, and treatment characteristics are shown in Table 1.

### 3.2. Subjective Assessment of EORTC QLQ-C30

#### 3.2.1. Global Health Status

All patients who completed the EORTC QLQ-C30 questionnaire before treatment were included. The completion compliance after treatment was 95% (92/97) immediately after treatment, 81% (75/93) after 3 months, and 68% (49/72) after 12 months. At 12 months after treatment, 10% (5/52) of patients who received curative treatment had died. Only one of these five patients was treated with superficial HT, and the other four patients were treated with deep regional HT. In total, 56% (25/45) of patients who were treated with palliative intent had died at 12 months. At three months after palliative treatment, 36% (16/45) of patients had already died, all having been treated with superficial HT. In total, 32 patients who received curative RCT with HT returned the EORTC QLQ-C30 questionnaire before treatment and also twelve months after treatment. The Global Health Status difference between these patients before and 12 months after treatment was not significant (74.99 ± 18.56 vs. 78.64 ± 15.69, *p*-value = 0.399), as shown in Figure 2. Similar results were obtained for the Global Health Status of 31 out of 45 patients who answered the Global Health status scale items and received palliative treatment before and three months after treatment (Figure 2, 59.94 ± 19.88 vs. 66.39 ± 21.02, *p*-value = 0.219). If patients treated with deep or superficial HT were analyzed separately, no difference in mean Global Health Status was found either before or at 3 or 12 months after treatment.

The mean Global Health Status score at baseline was higher for patients treated with curative intent than the EORTC general cancer population (75 ± 15.7 vs. 61.3 ± 24.2, *p* = 0.009). At 12 months after treatment, the relatively high mean Global Health Status score could be maintained in 32 patients treated with curative RCT and HT compared with the EORTC general cancer population (78.64 ± 18.56 vs. 61.3 ± 24.2, *p* = 0.002). No significant difference was found for the Global Health Status of 31 patients who received palliative treatment three months after treatment (66.39 ± 21.02 vs. 61.3 ± 24.2, *p* = 0.379) when compared with baseline and the EORTC reference population.

#### 3.2.2. Functional Scales

There were 32 patients treated with curative RCT and HT who returned the QoL at baseline and 12 months after treatment. Among the five functional scales, EF and SF items were significantly improved 12 months after treatment in patients treated with curative RCT and HT (Figure 3). The mean scores of the EF functional scale before and 12 months after treatment were 76.73 ± 21.91 and 90.36 ± 13.57, respectively (*p* = 0.0003), and the SF functional scales were 83.33 ± 20.73 vs. 94.79 ± 11.54, respectively (*p* = 0.008). None of the functional scale items was significantly improved or worsened for 30 patients treated with palliative RCT and HT, who did return the questionnaire after three months.

The subgroup analysis of different HT techniques showed that if the patients were treated with curative RT and superficial HT, the functional scale SF was significantly improved 12 months after treatment (85.96 ± 20.98 vs. 97.36 ± 8.35, *p* = 0.048) (Appendix A). Moreover, at 12 months, the subgroup of patients who received curative RCT and deep regional HT showed improvement in CF (92.30 ± 11.01 vs. 100 ± 0.0, *p* = 0.017) and EF (69.87 ± 23.20 vs. 91.66 ± 12.72, *p* = 0. 008) functional item scores (Appendix A). None of the functional scale items were significantly higher or lower for patients who received palliative RCT in combination with superficial and deep regional HT.

The functional scale items of 32 patients who received curative RCT combined with HT were compared with the EORTC general population. None of the functional scale mean scores for patients treated with curative RCT, except PF2 at baseline, were significantly different in comparison with the EORTC general cancer population. All functional scale items, except CF, were significantly improved 12 months after curative treatment (Appendix A). However, no difference in functional scale items was found between the EORTC general population and 30 patients treated with palliative RCT combined with HT before treatment and three months after treatment (Appendix A).

#### 3.2.3. Symptoms Scales

The nine items of the symptoms scale were evaluated for patients treated with curative and palliative RCT and HT.

The change in symptom scale scores before treatment and 12 months after treatment for patients who received curative RCT and HT was not significant, except for FI (*p* = 0.04) and SL (*p* = 0.01) (Figure 4). The analysis showed non-significant improvements in AP, CO, DI, DY, FA, NV, and PA symptom scale items (Figure 4). Patients who received palliative treatment had a modest decrease in the mean scores of the following symptom scale items after three months: CO, DI, FA, and SL and a significant reduction of FI (20 ± 25.67 vs. 10 ± 23.41, *p* = 0.04) and PA (45 ± 33.37 vs. 24.44 ± 25.04, *p* = 0.01), as shown in Figure 5. The AP, DY, and NV were symptom scale items that did not differ significantly from baseline three months after treatment, as illustrated in Figure 5.

There were no major differences in terms of symptom scale items prior to and 12 months after treatment for patients who received curative treatment using superficial and deep regional HT, respectively. Patients who received curative RCT in combination with deep HT had significant improvement only in the FI (17.94 ± 22 vs. 2.56 ± 9.24, *p* = 0.03) and SL (30.76 ± 25.31 vs. 7.69 ± 19.97, *p* = 0.01) functional scales (Appendix A). Moreover, the subgroup analysis performed in patients treated with palliative intent using superficial and deep regional HT also revealed no significant differences.

The symptom scale items of patients who received curative and palliative RCT combined with HT were compared with the EORTC general population. The comparison of mean scores at baseline (before treatment) in 32 and 30 patients treated with curative and palliative RCT combined with HT showed no significant difference with the EORTC general population. The symptom scale items such as AP, CO, FA, FI, NV, and SL were significantly improved 12 months after treatment (Appendix A). Positive outcomes were obtained after comparing the EORTC general population and 30 patients treated with palliative RCT combined with HT three months after treatment (Appendix A). With the exception of FA, the mean scores of other items were slightly lower than those of the EORTC general population, as shown in Appendix A.

### 3.3. Acute Toxicity Analysis

The summary of acute toxicity symptoms according to CTCAE (version 4.0) for patients treated with curative and palliative RCT and HT are summarized in Table 2 and Table 3, respectively.

The available medical records of 51 out of 52 patients treated with curative RCT in combination with HT indicated 88% grade 1, 44% grade 2 and 22% grade 3 toxicity symptoms during treatment (Table 2). During the HT sessions, one patient reported nausea/vomiting after anesthesia when inserting the temperature probe and tiredness after treatment. Additionally, another patient experienced stronger pain during HT. At three months after treatment, toxicity information for only 45 out of 52 patients treated with curative intent was available from the medical records. These reports showed that this group of patients had 23% grade 1, 2% grade 2 and 4% grade 3 symptoms (Table 2).

The acute toxicities were documented during the treatment and three months thereafter for 42 and 29 palliative patients, respectively (Table 3). During the treatment with RCT and HT, patients with palliative treatment intent had 85% grade 1, 14% grade 2 and 4% grade 3 toxicity symptoms (Table 3). Two patients reported an immediate and transient increase in pain during deep regional HT due to the lengthy duration of treatment (60 min). After three months, 29 patients treated with palliative intent had 44% grade 1, 7% grade 2, and none had grade 3 symptoms, as summarized in Table 3. After three months, only one patient had grade 4 toxicity, which was the development of hydronephrosis probably unrelated to HT.

### 3.4. Survival Outcomes

The 1-year OS rates for all patients treated with curative and palliative RCT combined with HT were 90% (95% CI: 79–96%) and 44% (95% CI: 31–59%), respectively (Figure 6).

For patients treated with superficial HT, the 1-year OS rates were 96% (95% CI: 82–99%) and 30% (95% CI: 13–48%) for patients treated with curative and palliative intent, respectively. Patients treated with regional deep HT had 1-year OS rates of 84% (95% CI: 65–94%) and 65% (95% CI: 45–81%) in patients with curative and palliative intent, respectively.

The 1-year LPFS rates were 94% (95% CI: 84–98%) and 64% (95% CI: 50–77%) for all patients treated with curative and palliative RCT and HT, respectively, as shown in Figure 7. For patients treated with either curative or palliative RT and superficial HT, the 1-year LPFS rates were 100% (95% CI: 88–100%) and 77% (95% CI: 57–90%), respectively. These rates for patients treated with curative and palliative RCT and deep HT were 88% (95% CI: 70–96%) and 52% (95% CI: 33–71%), respectively.

## 4. Discussion

This retrospective analysis provides real-world evidence regarding QoL in a population of cancer patients with different tumor entities treated with curative or palliative RCT combined with HT in a single institution. To the best of our knowledge, this is the first study to comprehensively analyze QoL using the EORTC QLQ-C30 questionnaire RCT in a real-world patient cohort. The analysis shows that the Global Health Status was not influenced by the combined treatment for both patient populations treated with curative or palliative intent. However, patients treated with curative RCT and HT experienced significant improvement in functional scales EF (*p* = 0.0003), SF (*p* = 0.02), FI (*p* = 0.04) and SL (*p* = 0.01) 12 months after treatment. In palliative patients, the symptom scale items FI (*p* = 0.04) and PA (*p* = 0.01) were significantly improved three months after palliative RCT and HT. Moreover, none of the other functional and symptoms scale items showed a tendency to worsen for patients treated with curative or palliative intention.

The subgroup analyses of different HT techniques showed that the mean value of the Global Health Status scale items did not significantly vary in patients who received curative treatment with RCT, including superficial or deep regional HT. Similar findings were obtained for the patients who were treated with palliative treatment with superficial or deep regional HT. With regard to functional scale items, patients treated with curative RT and superficial HT had a significantly increased mean SF value (*p* = 0.04), whereas patients treated with deep regional HT had significantly increased mean scores for CF (*p* = 0.01) and EF (*p* = 0.008). Patients who received curative RCT in combination with deep regional HT had significant improvements only in FI (*p* = 0.03) and SL (*p* = 0.01) symptom scale items.

The EORTC research group developed a widely used Patient-Reported Outcome Measures (PROMs) that assesses the quality of life of patients with cancer, known as EORTC QLQ-C30 [29]. Although the EORTC QLQ-C30 questionnaire is used as an outcome measurement in many cancer clinical trials and observational studies, its use in daily clinical practice has not been widely implemented. Jacobsen et al. concluded that the perceived barriers to incorporating QoL clinical routine assessments are inexperience with QoL assessment, skepticism about the sensitivity and specificity of QoL instruments, and logistic barriers [32]. In particular, the number and timing of QoL assessments, including the handling of missing data during the follow-up of patients, are major methodologic issues in the assessment of QoL endpoint [32]. Because the combination of hyperthermia and radiotherapy is usually offered to patients with very advanced cancers based on individual decisions and with rather outdated clinical trials, we decided to prospectively collect QoL data in all patients treated with either superficial or deep hyperthermia in our center.

Previous studies have demonstrated that patients treated with HT in combination with RCT generally have a good QoL [24,25,26]. However, these QoL analyses were only performed in small subgroups of patient cohorts. Gani et al. [24] reported that patients with locally advanced rectal cancer treated with RCT in combination with deep HT had significantly higher scores for the PF2 functional scale item (*p* = 0.02) and lower scores for the SF functional scale item (*p* < 0.0001) in comparison with the EORTC general population. Similarly, our analysis shows that patients treated with curative RCT in combination with HT, of which 12% (*n* = 6) were patients with rectal cancer, had not only significantly improved PF2 (*p* = 0.003) but also EF (*p* = 0.0003) and RF2 (*p* = 0.009) 12 months after treatment when compared with the EORTC general population. In the study of Gani et al. [24], the symptom scale items DI (*p* < 0.0001) and CO (*p* = 0.001) were worse for rectal cancer patients compared with the EORTC general population. In contrast, our real-world analysis of patients treated with curative intent showed significantly better, i.e., lower mean AP (*p* = 0.006), CO (*p* = 0.01), FA (*p* = 0.01), FI (*p* = 0.006), NV (*p* = 0.01) and SL (*p* = 0.009) scores. Additionally, the results of a planned interim analysis in patients with biochemically recurrent prostate cancer, who were treated with salvage RT and deep regional HT after prostatectomy, showed that there were no significant differences three months after treatment in any of the EORTC QLQ-C30 questionnaire items [25]. Similar results were obtained for patients with muscle-invasive bladder cancer who received RCT combined with regional deep HT after 12 months of follow-up [26]. The gastrointestinal quality of life index (GIQLI) questionnaire, which was specifically developed to measure QoL in patients suffering from malignant and benign diseases of the gastrointestinal tract and which correlates well with the EORTC QLQ-C30 [33,34], was used in patients with advanced rectal cancer who were treated with neoadjuvant RCT with or without HT [35]. In the latter study, no statistically significant differences in the global GIQLI index were reported between patients treated with neoadjuvant RCT with and without HT at any time point [35].

Our results confirm previous studies showing that HT does not substantially increase toxicity when combined with RCT in cancer patients if the temperature is controlled and kept below certain limits [3,14,36,37]. A prospective clinical study showed that if the maximum temperature achieved is greater than 45 °C, the rate of acute complications increases [38]. Similarly, a retrospective study showed that the incidence of acute skin reactions for patients treated with superficial HT combined with RT is significantly increased when the temperature applied exceeds 43 °C [39]. Additionally, the number of thermometry sensors and depth of treatment volume were associated with acute toxicity induced by HT in recurrent breast cancer [40]. Furthermore, a randomized study analysis showed that only 2.2% of 182 pelvic cancer patients treated with regional deep HT in combination with RT had grade III-IV acute toxicity [41]. Contrary to these results, Engin et al. showed that temperature parameters and acute skin reactions were not correlated in patients with superficial cancers [42]. Tilly et al. reported that T90, defined as ‘temperature achieved in 90% of the measured points”, and maximum temperature were not reported to correlate with acute or late toxicity [17]. Based on the above-mentioned studies, temperature monitoring during HT sessions to avoid any ‘hotspots’ above 43 °C is mandatory to avoid unnecessary skin toxicity [17,38,39]. The only and rare toxicity-related symptoms reported by patients during HT were pain, tiredness and nausea/vomiting due to the positioning and insertion of temperature measurement probes.

As expected, patients treated with curative intent had higher 1-year OS rates in comparison with patients treated with palliative intent. The palliative treatment was intended to achieve adequate pain and symptom control during a life expectancy of usually only a few months. Consequently, a clinically relevant time point of three months after treatment was chosen for the QoL assessment for this group of patients. It has been shown that patients with different cancer diagnoses and stages treated with RT only may experience a deterioration in QoL, which recover a month after RT [43]. Our results are in line with this and indicate that treatment including HT does not lead to a sustained impairment in QoL for palliative patients.

The main limitations of this study are the retrospective data analysis of patients with a high variability of tumor sites. In addition, not all patients returned the QoL questionnaires during follow-up, leading to variability in the number of patients that could be included in the analysis at different time points after treatment. At 12 months after treatment, there were 72 patients alive, and 49 out of 72 returned the QoL. This might introduce an upward bias in the analysis of patients treated with curative intent. The main strength of this study is that it is a real-world analysis of QoL in all patients treated with RCT in combination with HT in one institution and during a defined period. Thus far, no published study has systematically focused on such a real-world patient population to investigate whether HT might induce side effects that adversely affect the QoL of cancer patients. We propose that integrating the collection of short- and long-term patient QoL data into the clinical treatment workflow would help to evaluate the short- and long-term effects of HT in specific groups of cancer patients.

## 5. Conclusions

The use of superficial or deep regional HT in combination with RCT is a treatment strategy to improve outcomes and relieve the symptoms and, thus, to reduce the suffering caused by cancer. In this retrospective analysis, we observed that the Global Health Status of cancer patients could be maintained after palliative or curative treatment with RCT in combination with HT. Moreover, a significant improvement of functional scale items EF (*p* = 0.0003), SF (*p* = 0.02), FI (*p* = 0.04) and SL (*p* = 0.01) 12 months after treatment was found in patients treated with curative intent in this analysis. The symptom scale items FI (*p* = 0.04) and PA(*p* = 0.01) were the only QoL parameters that were significantly improved three months after treatment in patients RCT treated with palliative intent. None of the EORTC QLQ-C30 scale items showed significant worsening of QoL. Further research is required to better understand the QoL of patients who received HT when combined with RCT.

## Figures and Tables

**Figure 1 cancers-15-01241-f001:**
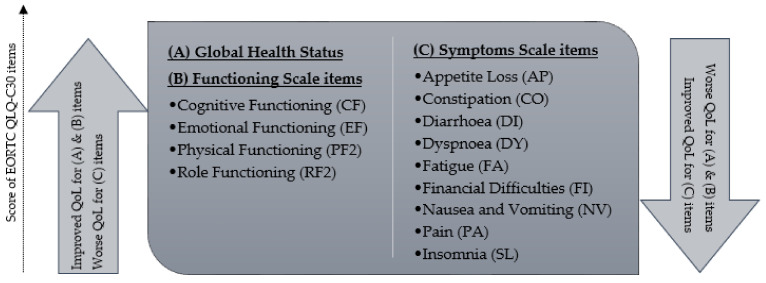
The items of the EORTC QLQ-C30 scale.

**Figure 2 cancers-15-01241-f002:**
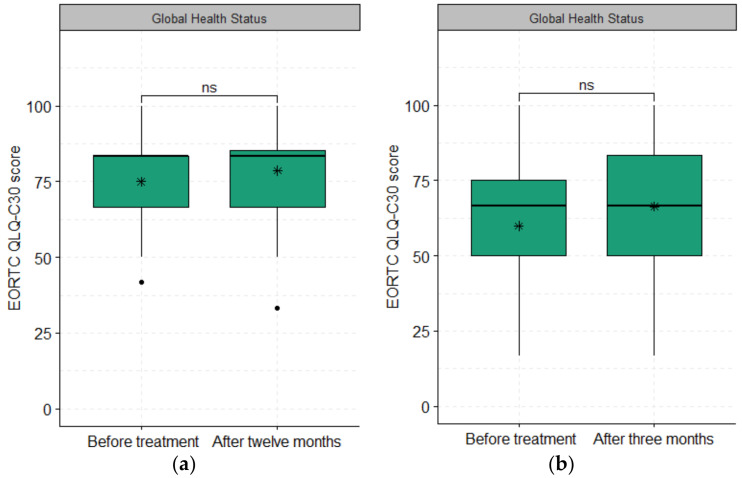
The global health status scores of (**a**) 32 patients treated with curative RCT and HT before and 12 months after treatment (74.99 ± 18.56 vs. 78.64 ± 15.69, *p*-value = 0.245) and (**b**) 31 patients treated with palliative RCT and HT before and three months after treatment (59.94 ± 19.88 vs. 66.39 ± 21.02, *p*-value = 0.227). The asterisks (*) represent the mean values; ns: non-significant with *p*-value > 0.05.

**Figure 3 cancers-15-01241-f003:**
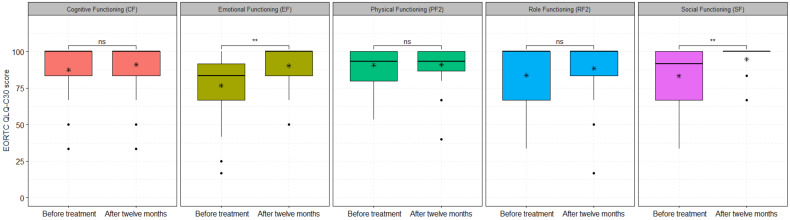
The functional scale scores of 32 patients treated with curative RCT and HT before and 12 months after treatment for: CF (87.50 ± 16.93 vs. 91.14 ± 16.38, *p*-value = 0.23), EF (76.73 ± 21.91 vs. 90.36 ± 13.57, *p*-value = 0.003), PF2 (90.83 ± 11.57 vs. 91.04 ± 12.51, *p*-value = 0.9), RF2 (83.85 ± 21.37 vs. 88.54 ± 19.13, *p*-value = 0.36), and SF (83.33 ± 20.73 vs. 94.79 ± 11.54, *p* = 0.008). The asterisks (*) represent the mean values; ns: non-significant with *p*-value > 0.05; **: significant with *p*-value < 0.01.

**Figure 4 cancers-15-01241-f004:**
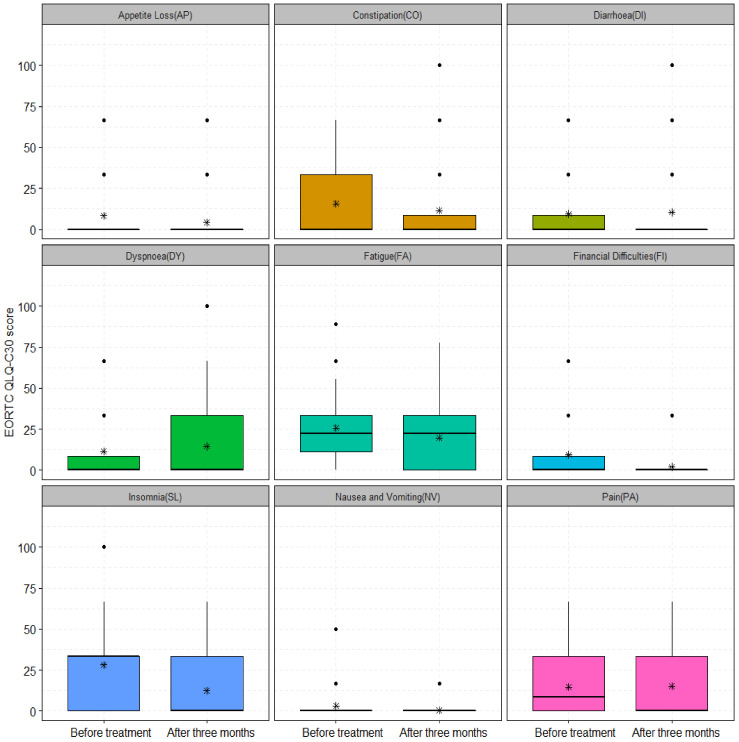
The symptom scale scores of 32 patients treated with curative RCT and HT before and 12 months after treatment for: AP (8.33 ± 16.93 vs. 4.16 ± 14.04, *p*-value = 0.19), CO (15.62 ± 20.71 vs. 11.45 ± 23.35, *p*-value = 0.23), DI (9.37 ± 20.71 vs. 10.41 ± 23.09, *p*-value = 0.85), DY (11.45 ± 21.76 vs. 14.58 ± 26.69, *p*-value = 0.74), FA (25.69 ± 22.12 vs. 19.79 ± 19.49, *p*-value = 0.31), FI (20 ± 25.67 vs. 10 ± 23.41, *p*-value = 0.04), NV (3.12 ± 9.87 vs. 0.52 ± 2.94, *p*-value = 0.17), PA (14.58 ± 17.83 vs. 15.1 1 ± 20.89, *p*-value = 0.84), and SL (28.12 ± 26.92 vs. 12.50 ± 20.31, *p*-value = 0.01). The asterisks (*) represent the mean values; ns: non-significant with *p*-value > 0.05; *: significant with *p*-value < 0.05.

**Figure 5 cancers-15-01241-f005:**
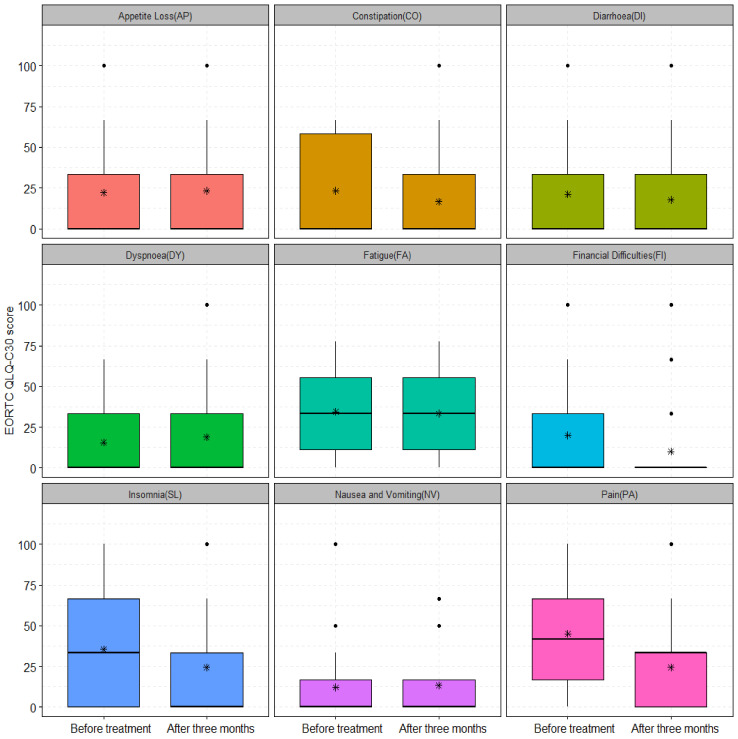
The symptom scale scores of 30 patients treated with palliative RCT and HT before and three months after treatment for: AP (22.22 ± 31.96 vs. 23.33 ± 32.92, *p*-value = 0.94), CO (23.33 ± 29.23 vs. 16.66 ± 28.71, *p*-value = 0.31), DI (21.11 ± 29.23 vs. 17.77 ± 29.98, *p*-value = 0.39), DY (15.55 ± 20.96 vs. 18.88 ± 24.26, *p*-value = 0.63), FA (34.44 ± 23.94 vs. 33.33 ± 24.41, *p*-value = 0.86), FI (20 ± 25.67 vs. 10 ± 23.41, *p*-value = 0.04), NV (12.22 ± 23.13 vs. 13.33 ± 23.73, *p*-value = 0.92), PA (45 ± 33.37 vs. 24.44 ± 25.04, *p*-value = 0.01), and SL (35.55 ± 31.48 vs. 24.44 ± 31.48, *p*-value = 0.14). The asterisks (*) represent the mean values; ns: non-significant with *p*-value > 0.05; *: significant with *p*-value < 0.05.

**Figure 6 cancers-15-01241-f006:**
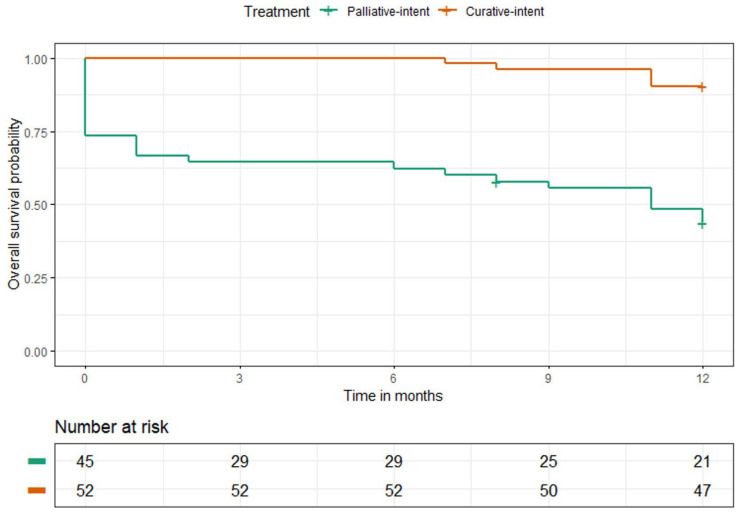
One-year OS probability of all patients treated with palliative and curative RCT combined with HT.

**Figure 7 cancers-15-01241-f007:**
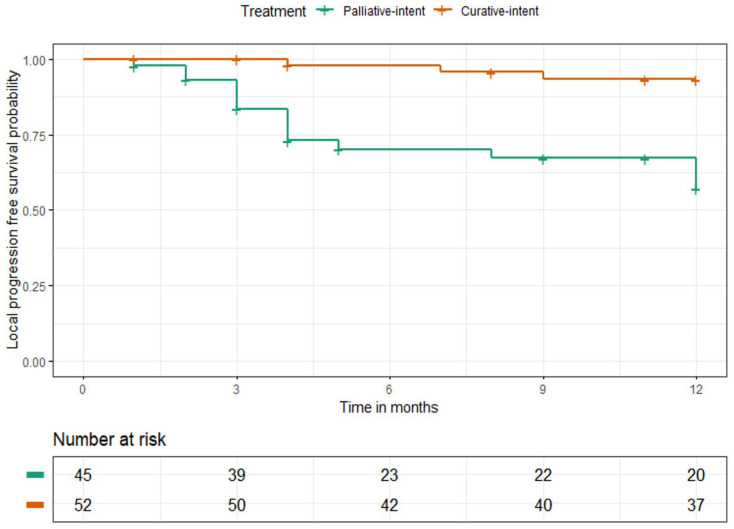
One-year LPFS probability of all patients treated with palliative and curative ra-dio(chemo)therapy combined with HT.

**Table 1 cancers-15-01241-t001:** Patient, disease, and treatment characteristics.

Variables	Palliative Intent Treatment, *n* (%)	Curative Intent Treatment, *n* (%)
**Sex**		
Male	25 (44)	17 (33)
Female	20 (56)	35 (67)
**Median age at treatment start** ^‡^	66 (37–92)	66 (28–84)
**KPS prior to treatment**		
100	14 (31)	25 (48)
90	7 (16)	6 (12)
≤80	14 (31)	5 (10)
n.a.	10 (22)	16 (30)
**Primary tumor histology**		
Breast cancer	11 (25)	22 (42)
Lung cancer	5 (11)	0 (0)
Bladder cancer	2 (4)	7 (13)
Anal cancer	4 (9)	2 (4)
Rectal cancer	4 (9)	6 (11)
Melanoma	6 (13)	0 (0)
Sarcoma	1 (2)	5 (10)
Pancreas cancer	0 (0)	5 (10)
Prostate cancer	5 (11)	2 (4)
Other	7 (16) *	3 (6) **
**Recurrent cancer**	25 (56)	32 (62)
**RT prescribed total dose (Gy)** ^†^	38 (±10)	50 (±6)
**RT dose per fraction (Gy)** ^†^	2.5 (±0.5)	1.9 (±0.2)
**Number of RT fractions** ^†^	16 (±7)	24 (±4)
**Duration of RT + HT (days)** ^†^	25 (±13)	35 (±8)
**HT techniques**		
Superficial HT	22 (49)	27 (52)
Deep regional HT	23 (51)	25 (48)
**Time interval between RT and HT (minutes)** ^‡^	15 (7–80)	14 (6–83)
**n.a.**	3 (7)	7 (13)
**Total HT sessions** ^‡^	5 (3–11)	5 (4–11)
**Temperature metrics (°C)**		
T_avg_ ^†^	39.9 (±0.7)	40.3 (±0.6)
T_max_ ^†^	40.5 (±0.8)	40.9 (±0.6)
n.a.	11 (32)	10 (19)
**Thermal dose CEM43** ^†^	1.7 (±1.5)	2.8 (±2.2)
n.a.	11 (32)	10 (19)
**Chemotherapy**		
Superficial HT	0 (0)	0 (0)
Deep regional HT	1 (4)	14 (27)

^‡^: median (range); ^†^: mean (± standard deviation); n.a.: not available; *: cervical (*n* = 1), gastroesophageal cancer (*n* = 1), liver cancer (*n* = 2), parotid glad (*n* = 1), renal cell carcinoma (*n* = 1), thyroid (*n* = 1); **: bone cancer (*n* = 1), cervical cancer (*n* = 1), hypopharyngeal cancer (*n* = 1).

**Table 2 cancers-15-01241-t002:** Acute toxicities of patients treated with curative intent.

	Toxicity	Grade 1	Grade 2	Grade 3
		*n* (%)	*n* (%)	*n* (%)
During treatment, *n* = 51	Dermatitis	7 (14)	7 (14)	3 (6)
	Mucositis	0	0	1 (2)
	Skin reaction(s)	7 (14)	6 (12)	4 (8)
	Diarrhoea	8 (16)	2 (4)	2 (4)
	Urinary incontinence/ urinary retention/ Nocturia	4 (8)	0	0
	Dysuria	3 (6)	1 (2)	0
	Pyelonephritis	0	1 (2)	0
	Nausea	1 (2)	1 (2)	0
	Vomiting	1 (2)	0	0
	Constipation	0	2 (4)	0
	Pain	1 (2)	0	1 (2)
	Tiredness/fatigue	11 (22)	2 (4)	0
	Appetite loss	1 (2)	0	0
Three months after treatment, *n* = 45	Skin reaction(s)	4 (9)	0	0
	Diarrhoea	3 (6)	0	0
	Urinary incontinence/ urinary retention	2 (4)	0	0
	Dysphagia	1 (2)	0	0
	Nausea	0	0	1 (2)
	Pain	1 (2)	1 (2)	0
	Vomiting	0	0	1 (2)

**Table 3 cancers-15-01241-t003:** Acute toxicities of patients treated with palliative intent.

	Toxicity	Grade 1	Grade 2	Grade 3
		*n* (%)	*n* (%)	*n* (%)
During treatment *n* = 42	Dermatitis	4 (9)	3 (7)	0
	Skin reaction(s)	4 (9)	5 (11)	0
	Hepatitis	0	1 (2)	1 (2)
	Urinary incontinence/ urinary retention	5 (11)	0	0
	Diarrhoea	7 (16)	0	1 (2)
	Dysphasia	1 (2)	0	0
	Pain	7 (17)	0	0
	Tiredness/fatigue	9 (21)	1 (2)	0
Three months after treatment *n* = 29	Dermatitis	1 (3)	0	0
	Skin reaction(s)	3 (10)	0	0
	Diarrhoea	2 (7)	0	0
	Urinary incontinence/ urinary retention	4 (14)	0	0
	Pain	3 (10)	2 (7)	0

## Data Availability

The data presented in this study are available on request from the corresponding author. The data are not publicly available due to privacy and ethical reasons.

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
