# Peer review of "Real World Analysis of Quality of Life and Toxicity in Cancer Patients Treated with Hyperthermia Combined with Radio(chemo)therapy"

_cancers, 2023, doi:10.3390/cancers15041241_

Round 1

Reviewer 1 Report

Thanks for inviting me to review this work.

I find it interesting to study the QoL of cancer patients. 

Everything seems to be sound and fine. 

Author Response

General comments:

"Thanks for inviting me to review this work. I find it interesting to study the QoL of cancer patients. Everything seems to be sound and fine."

Answer to reviewer 1: We thank the reviewer for the general comment. There was no question raised by the Reviewer #1.

Reviewer 2 Report

Hyperthermia can be added to radiotherapy and chemotherapy in the treatment of malignant cancers. Clinical results have been shown that adding this treatment modality to radiotherapy and chemotherapy improves treatment outcome, without severe toxicity. However, no qualitative parameters on quality of life have been investigated. The authors provide an assessment of the quality of life (following the EORTC QLQ-C30) of patients treated with radiotherapy and/or chemotherapy combined with hyperthermia. Two groups are distinguished, patients treated with curative and palliative intent. Moreover, acute toxicity as well as, overall survival and local progression free survival (one year) are also reported.

The authors should provide more information on how qualitative analysis based on questionnaires ca be performed, and what are the difficulties and problems that these type of studies can face.

What do the authors mean with “real world analysis”, “real world conditions”, “real world patients”? Are there no real world analysis or patients? Please consider explaining or rephrase the mentioned sentences.

Many similar figures are presented (both on manuscript and supplementary materials) in which comparisons are not significant. Message could be clear if focused more on the main findings and avoiding including too many plots from which results can be included in text.  

Specific comments

Introduction

Line 55: Statement about correlation between temperatures and thermal toxicity. Please verify references, because many studies have been shown the opposite, specifically for acute side effects (Seegenschmiedt et al., 1989; Engin et al., 1993; Linthorst et al., 2013; Bakker et al., 2019, etc).

Line 73: “Real world conditions” is mentioned. Please clarify the meaning.

Methods

Lines 88-89:  Are both superficial and deep treatments applied for one hour (steady state) after heating up phase? Please consider a separate description for superficial and deep hyperthermia treatments providing more detail.

Lines 89-90 is the range 6 to 1127min correct. Please verify (1127min = >18hours?)

Line 92: Missing closing bracket after brand of applicator.

Line 99: Please specify why two different applicators are used (Sigma-60 and Sigma-Eye).

Line 101: Treatment time specified again, delete previous part in lines 88-89. Please consider specifying for superficial with more detail in previous paragraph.

Line 105: Mapping was performed every 10 seconds? Please specify the acquisition method (e.g., probe moved every x cm along a track of up to y cm with repetitions every 10 seconds?).

Lines 121: For the description of the Questionnaire and the different scales, a summary illustrating image would be a good addiction.  

Line 140: Please provide a reference for the criteria used to evaluate toxicity.

Line 141: Please specify the time of the overall survival analysis and local progression free survival. Please also indicate the time point from which it was calculated, meaning, for example, from the last hyperthermia treatment or the point of diagnosis…

Results

Lines 159-162: Information given is not specified in the table. Please consider mentioning the number of superficial and deep treatments per group of patients in the table. The number of patients receiving chemotherapy should be mentioned for all types of hyperthermia and match the information in the table. Please verify.

Line 172: Please specify what was the objective of the questionnaire immediately after treatment finish.

Line 173: Please add to the number of patients who died with palliative the total number in the beginning (25/52).

Line 174: Please verify that the patient numbers are correct. In line 158 the total number of curative patients included is 45 and not 52 as stated in this part.

Line 179-183: Why only 32 patients out of the initial 45 treated with curative intent. Please provide an explanation. Same applies to the 31/52 treated with palliative intent. Please verify the p-values on the text because they do not match the values in Figure 1.

Figure 1. Please consider changing the plot to a more readable box plot, since this figure is not intuitive and difficult to understand.

Line 202: P-value of SF does not match the one in caption of Figure 2.

Line 203: Why only 30/52 patients? Compared to also the 31 mentioned in the previous section. Please explain. Consider deleting Figure 3 since the results are not significant.

Line 207 and Figure S1: P-values of SF not equal.

Line 209 and Figure S2: P-values of SF not equal. Does it make sense to compare when only one value is available in the case of CS? Why are not there 19 values? Please be consistent using NS or ns for non-significant (Figure S2).

Consider removing Figures S3 and S4.

Line 233: Again 30 patients for palliative (31?).

Line 242: The significance for FI comes from the comparison of only two point in the group after treatment. Is that really significant? Does that provide sufficient evidence?

Line 245: Same question as above. The significance for FI comes from the comparison of only two point in the group after treatment. Is that really significant? Does that provide sufficient evidence?

Lines 251-256: Consider removing Figures S5, S7 and S8. In Figure S6, be consistent with ns (or NS). Significance in FI and SL scales also come from 2 or 3 values comparison in the after twelve moths group.

Line 291: 51 patients treated with curative intent (total of 45 reported in the beginning of the results section-line158). Please verify and explain if different. Please explain why after three months the total number of patients is different.

Line 299: Did you monitor any skin burns or subcutaneous burns, fat necrosis?

Line 301: 42 patients treated with curative intent (total of 52 reported in the beginning of the results section-line158). Please verify and explain if different. Please explain why after three months the total number of patients is different.

Line 313/Figure 6: Please verify number of total patients and compare to line 158. What is the objective of the plot, if the overall survival of the two groups is not aimed to be compared?

Line 324/Figure 7: Please verify number of total patients and compare to line 158. What is the objective of the plot, if the local progression free survival of the two groups is not aimed to be compared?

Figures S9 and S10: Could include a comparison between deep and superficial for the same group and present if significand different in the plot. Otherwise, consider remove, since it does not provide more information than the values presented in the text.

Discussion

Line 336/340/401: Please explain what is real world evidence. quality of life analysis is not always a real world measure? Are there studies on theoretical quality of life?

Line 396: The number of patients was mentioned to be the innovative part of the presented study. Please be consistent and try to better explain.

Please comment the statistical significance of tests that are performed with groups with only two or three patients and how that influences the results and conclusions of the current study. Consider also adding some information on how this type of analysis could be  performed with more patient compliance

Author Response

Summary: "Hyperthermia can be added to radiotherapy and chemotherapy in the treatment of malignant cancers. Clinical results have been shown that adding this treatment modality to radiotherapy and chemotherapy improves treatment outcome, without severe toxicity. However, no qualitative parameters on quality of life have been investigated. The authors provide an assessment of the quality of life (following the EORTC QLQ-C30) of patients treated with radiotherapy and/or chemotherapy combined with hyperthermia. Two groups are distinguished, patients treated with curative and palliative intent. Moreover, acute toxicity as well as, overall survival and local progression free survival (one year) are also reported."

General comments:

  1. "The authors should provide more information on how qualitative analysis based on questionnaires can be performed, and what are the difficulties and problems that these type of studies can face."

We thank the reviewer very much for this suggestion and introduced the following sentences in the discussion part:

Line 455-467 in the revised manuscript: The EORTC research group developed a widely used Patient-Reported Outcome Measures (PROMs), that assesses the quality of life of patients with cancer, known as EORTC QLQ-C30 [29]. Although the EORTC QLQ-C30 questionnaire is used as an outcome measurement in many cancer clinical trials and observational studies, its use in daily clinical practice has not been widely implemented. Jacobsen et al. concluded that the perceived barriers to incorporating QoL clinical routine assessments are inexperience with QoL assessment, skepticism about sensitivity and specificity of QoL instruments, and logistic barriers [32]. In particular, the number and timing of QoL assessments including the handling of missing data during the follow up of patients are major methodologic issues in the assessment of QoL endpoint [32]. Because the combination of hyperthermia and radiotherapy is usually offered to patients with very advanced cancers based on individual decision and with rather outdated clinical trials, we decided to prospectively collect QoL data in all patients treated with either superficial or deep hyperthermia in our center.

  1. What do the authors mean with “real world analysis”, “real world conditions”, “real world patients”? Are there no real world analysis or patients? Please consider explaining or rephrase the mentioned sentences.

The words "real world analysis", “real world conditions”, “real world patients” are used to refer that these patients are not a specific group of patients who were treated according to a standardized treatment standard or a treatment protocol. These analysis were not performed within a randomized clinical trial or prospective observational study. The patient cohort included in this study are all patients treated with hyperthermia and radio(chemo)therapy in our radiation oncology center without any selection. The real world analysis has gained increased  interest in recent years as summarized by Di Maio et al. 2020 (Real-World Evidence in Oncology: Opportunities and Limitations. The oncologist25(5), e746–e752)

With regard to reviewer's suggest to explain the real world conditions, we added the following in the Introduction paragraph:

Line 74-78 in the revised manuscript: "The term "real-world conditions" refers to a patient cohort where all patients treated with hyperthermia in our center were prospectively included and data were routinely collected from a variety of sources. The benefits and limitations of real world analyses in comparision to randomized clinical trials have been summarized elsewhere[27]. "

  1. Many similar figures are presented (both on manuscript and supplementary materials) in which comparisons are not significant. Message could be clear if focused more on the main findings and avoiding including too many plots from which results can be included in text.

The suggestion of reviewer is very much welcomed to remove the figures, which shows non-significant outcomes. Thus, we deleted the following figures, which did not present significant results:

  • Figure S3. The functional scale scores of 14 patients treated with palliative RCT and superficial HT before and three months after treatment;
  • Figure S4. The functional scale scores of 18 patients treated with palliative RCT and deep regional HT before and three months after treatment;
  • Figure S5. The symptom scale scores of 19 patients treated with curative RCT and superficial HT before and twelve months after treatment;
  • Figure S7. The symptom scale scores of 14 patients treated with palliative RCT and superficial HT before and three months after treatment;
  • Figure S8. The symptom scale scores of 16 patients treated with palliative RCT and deep regional HT before and three months after treatment;
  • Figure S9. 1-year OS probability of patients treated with curative and palliative RT combined with superficial HT and patients treated with curative and palliative RCT combined with deep HT;
  • Figure S10. 1-year LPFS probability of patients treated with curative and palliative RT combined with superficial HT and patients treated with palliative and curative RCT combined with deep HT.

Specific comments:

  1. "Introduction:

Line 55: Statement about correlation between temperatures and thermal toxicity. Please verify references, because many studies have been shown the opposite, specifically for acute side effects (Seegenschmiedt et al., 1989; Engin et al., 1993; Linthorst et al., 2013; Bakker et al., 2019, etc).

We appreciate the recommendation of reviewer for these important clinical studies which report the relationship of temperatures achieved during hyperthermia sessions and toxicity. We added the following paragraph in the discussion to elaborate more about the toxicity induced by superficial and deep regional hyperthermia:

Line 502-516 in the revised manuscript: A prospective clinical study showed that if the maximum temperature achieved is greater than 45°C, the rate of acute complications increased [38]. Similarly, a retrospective study showed that the incidence of acute skin reactions for patients treated with superficial HT combined with RT is significantly increased when the temperature applied exceeds 43°C [39]. Additionally, the number of thermometry sensors and depth of treatment volume were associated with acute toxicity induced by HT in recurrent breast cancer [40]. Furthermore, a randomized study analysis showed that only 2.2% of 182 pelvic cancer patients, treated with regional deep HT in combination with RT, had grade III-IV acute toxicity [41]. Contrary to these results, Engin et al. showed that temperature parameters and acute skin reactions were not correlated in patients with superficial cancers [42]. Tilly et al. reported that T90, defined as ‘temperature achieved in 90% of the measured points", and maximum temperature were not reported to correlate with acute or late toxicity [17]. Based on the above-mentioned studies, temperature monitoring during HT sessions to avoid any 'hotspots' above 43°C is mandatory to avoid unnecessary skin toxicity [17,38,39].

Line 73: “Real world conditions” is mentioned. Please clarify the meaning."

With regard to the reviewer's suggestion to explain the term real world conditions, we added the following in the introduction:

Line 74-78 in the revised manuscript: "The term "real-world conditions" refers to a patient cohort where.... "

  1. Methods
  • Lines 88-89:  Are both superficial and deep treatments applied for one hour (steady state) after heating up phase? Please consider a separate description for superficial and deep hyperthermia treatments providing more detail.

We thank very much the reviewer for the comment to include separate descriptions for superficial and deep regional hyperthermia treatments. We revised the whole section "2.2 Hyperthermia treatment" to provide a clearer and in detail description for superficial and deep hyperthermia treatments, as follows:

Line 88-128 in the revised manuscript:

2.2. Hyperthermia treatment

Superficial and deep regional HT were performed according to European Society for Hyperthermic Oncology (ESHO) guidelines [9,28]. For all patients treated with superficial and deep regional HT, the temperature metrics such as average temperature achieved in measurement points (Tavg) and maximum temperature achieved at measurement points (Tmax) were assessed. In addition, thermal dose CEM43 was calculated including the time interval between start of HT sessions and RT treatment.

2.2.1 Superficial hyperthermia treatment

Briefly, superficialSuperficial HT was administered using a BSD 500 applicator (BSD Medical Cooperation/Pyrexar, Salt Lake City, UT, USA) at a frequency of 915 MHz. Depending on the size of the superficial cancer, a surface applicator with 3, 8 or 24 spiral antennas covering diameters of 9 cm, 12 cm or an area of 24 x 20 cm was selected. The temperature probes were placed between the skin and the water bolus (filled with deionized water) to monitor the temperature on the surface of the target. After a 15-30 minute preheating phase, a therapeutic temperature between 40 and 43°C was applied for one hour. Patients received superficial HT sessions once or twice per week with a median time interval of 14 (range: 6-87) minutes before RT.

2.2.2 Deep regional hyperthermia treatment

For the treatment of deeply seated cancers, a deep regional HT technique was applied using the BSD 2000/3D system combined with the Sigma-60 or Sigma-Eye phased array applicator (BSD Medical Cooperation/Pyrexar, Salt Lake City, UT, USA) at the frequency range of 75 -120 MHz. The Sigma-60 and Sigma-Eye are annular phased array applicators that are comprised of a clear plastic shell, eight radiating dipoles, and a bolus membrane. The difference between Sigma-60 and Sigma-Eye applicators is the shape of the applicators' plastic shell, as Sigma-60 and Sigma-Eye applicators have cylindrical and elliptical shapes, respectively. The choice of applicators was based on the patient's size. Typically, the Sigma-Eye was used for smaller sized patients.

For thermometry, temperature probes were placed in the rectum or bladder or vagina (for female patients) and on the anal margin. The temperature was measured at 10-second intervals, starting before treatment and stopping immediately or five minutes after switching off the radiofrequency power. Thermal mapping was performed every 5 minutes with a step size of 1 cm and a maximum map length of 18 cm. After a 15-30 minute preheating phase, a therapeutic temperature between 40 and 43°C was applied for one hour. Patients were treated with deep regional HT sessions once or twice per week with a median time interval between of 16 (range: 11-83) minutes before RT.

  • Lines 89-90 is the range 6 to 1127min correct. Please verify (1127min = >18hours?)

The suggestion of reviewer to verify the results is very appreciated. The "1127 minutes" time interval was a technical error. We corrected this to "6 to 87 min (line 128 in the revised manuscript).

  • Line 92: Missing closing bracket after brand of applicator.

Thank you very much to the reviewer about the comment to add  the closing bracket.  We added the closing bracket:

Line 101 in the revised manuscript: "…(BSD Medical Cooperation/Pyrexar, Salt Lake City, UT, USA)…"

  • Line 99: Please specify why two different applicators are used (Sigma-60 and Sigma-Eye).

We added the following explanation:

Line 115-120 in the revised manuscript: "The Sigma-60 and Sigma-Eye are annular phased array applicators that are comprised of a clear plastic shell, eight radiating dipoles, and a bolus membrane. The difference between Sigma-60 and Sigma-Eye applicators is the shape of the applicators' plastic shell, as Sigma-60 and Sigma-Eye applicators have cylindrical and elliptical shapes, respectively. The choice of applicators was based on the patient's size. Typically, the Sigma-Eye was used for smaller sized patients."

  • Line 101: Treatment time specified again, delete previous part in lines 88-89. Please consider specifying for superficial with more detail in previous paragraph.

The treatment time including further specific information about superficial HT treatment are summarized in a separate subsection "2.2.1 Superficial hyperthermia treatment": Line 107-110 in the revised manuscript.

  • Line 105: Mapping was performed every 10 seconds? Please specify the acquisition method (e.g., probe moved every x cm along a track of up to y cm with repetitions every 10 seconds?).

We added the following explanation, line 124-128 in the revised manuscript:

"Thermal mapping was performed every 5 minutes with a step size of 1 cm and a maximum map length of 18 cm. After a 15-30 minute preheating phase, a therapeutic temperature between 40 and 43°C was applied for one hour. Patients were treated with deep regional HT sessions once or twice per week with a median time interval between of 16 (range: 11-83) minutes before RT."

  • Lines 121: For the description of the Questionnaire and the different scales, a summary illustrating image would be a good addiction.  

We agree with the reviewer comment and added an image, which illustrates the scale items of the EORTC QLQ-C30 questionnaire including the interpretation of scores for each scale item.

Therefore, we added Figure 1, which graphically represent all the items of EORTC QLQ-C30's scales.

  • Line 140: Please provide a reference for the criteria used to evaluate toxicity.

We agree with the reviewer that this is important. However, the reviewer might have missed the information that is already given in the methods part, line 174-177 in the revised manuscript:

"The acute toxicity symptoms were also collected from medical reports for each individual patient during treatment and three months after treatment. The toxicity symptoms were evaluated using the Common Terminology Criteria for Adverse Events (CTCAE) version 4.0."

  • Line 141: Please specify the time of the overall survival analysis and local progression free survival. Please also indicate the time point from which it was calculated, meaning, for example, from the last hyperthermia treatment or the point of diagnosis…

We thank the reviewer to provide the definitions of overall survival and progression free survival outcomes. We totally agree with the reviewer about the importance to provide the definitions, therefore we added the definitions including the censored events, line 177-1813 in the revised manuscript:

"LPFS was calculated between the start of last fraction of RT and the first of the following events: locoregional or metastatic recurrence from any cause, whereas OS was defined as the time period between the last fraction of RT and the date of death from any cause. Events were censored if information on time to event was not available due to loss to follow up, or if the outcome even did not occur before one year."

  1. Results
  • Lines 159-162: Information given is not specified in the table. Please consider mentioning the number of superficial and deep treatments per group of patients in the table. The number of patients receiving chemotherapy should be mentioned for all types of hyperthermia and match the information in the table. Please verify.

We are thankful to reviewer for emphasizing the importance to include further information in Table 1. Thus we added the information and updated Table 1 as suggested from the reviewer.

  • Line 172: Please specify what was the objective of the questionnaire immediately after treatment finish.

At the end of combined treatment with RT and HT, acute toxicity usually is at the highest level. Therefore, from the radiation oncology perspective, this is a very important timepoint for assessment of toxicity.

  • Line 173: Please add to the number of patients who died with palliative the total number in the beginning (25/52).

The total number of patients treated with palliative treatment were 45 and 52 with curative treatment. To avoid any confusion, we rephrased and added the following information in Section "3.2.1 Global Health Status", line 218-223 in the revised manuscript:

" At 12 months after treatment, 10% (5/52) of patients who received curative treatment had died. Only one of these five patients was treated with superficial HT and the other four patients were treated with deep regional HT. In total, 56% (25/45) of patients who were treated with palliative intent had died at 12 months. At three months after palliative treatment, 36% (16/45) of patients had already died, all having been treated with superficial HT."

  • Line 174: Please verify that the patient numbers are correct. In line 158 the total number of curative patients included is 45 and not 52 as stated in this part.

We confirm that the included patients treated with palliative treatment in the analysis are 45 and curative are 52. This is consistent with the information in Table 1.

  • Line 179-183: Why only 32 patients out of the initial 45 treated with curative intent. Please provide an explanation. Same applies to the 31/52 treated with palliative intent. Please verify the p-values on the text because they do not match the values in Figure 1.

As we stated in the section of "2.4. Statistical analysis" that "The missing data were excluded from the analysis and the comparison of EORTC QLQ-C30 items scores between two different time points was performed in the same group of patients." indicating that patients who did not return their questionnaire three months after treatment for palliative treatment and 12 months for curative treatment were excluded also from the baseline to avoid any bias. To clarify this for the reader we added the following sentences, line 229-230 in the revised manuscript:

"In total, 32 patients who received curative RCT with HT returned the EORTC QLQ-C30 questionnaire before treatment and also twelve months after treatment. "

We also added the following sentence, 233-234 in the revised manuscript:

"Similar results were obtained for Global Health Status of 31 out of 45 patients who answered the Global Health status scale items and "

  • Figure 1. Please consider changing the plot to a more readable box plot, since this figure is not intuitive and difficult to understand.

With the previous recommendation of the reviewer to change the type of the plot of Figure 1 (in the revised manuscript labelled as Figure 2) in a way that the figure becomes easier to interpret for the reader, now  two boxplots were created to show the difference of Global Health status for patients treated with curative and palliative treatment at 12 months and three months, respectively.

  • Line 202: P-value of SF does not match the one in caption of Figure 2.

Line 262 in the revised manuscript: We added the correct p-value for the SF item in the text.

  • Line 203: Why only 30/52 patients? Compared to also the 31 mentioned in the previous section. Please explain. Consider deleting Figure 3 since the results are not significant.

We thank the reviewer for emphasizing the need to explain the variation of number of patients. Therefore, we add the following information in the beginning of the Section "3.2.2. Functional scales":

Line 258-259 in the revised manuscript:

"There were 32 patients treated with curative RCT and HT who returned the QoL at baseline and 12 months after treatment."

and line 265 in the revised manuscript

"…, who did return the questionnaire after three months."

  • Line 207 and Figure S1: P-values of SF not equal.

It is very much appreciated that the reviewer pointed out that the last digit of p-value was missing. Line 267 in the revised manuscript: we corrected this mistake.

  • Line 209 and Figure S2: P-values of SF not equal. Does it make sense to compare when only one value is available in the case of CS? Why are not there 19 values? Please be consistent using NS or ns for non-significant (Figure S2).

The comments of the reviewer are very much appreciated. We would like to provide the further explanation:

As described in the Section "2.4 Statistical analysis ", the change score between baseline and at a different timepoint was assessed using the same group of patients. This means that the EORTC QLQ-C30 questionnaires of 13 patients treated with curative RCT and deep regional HT were used to evaluate the functional scale scores before treatment and 12 months after treatment as shown in Figure S2.

In Figure S2, the boxplots show all functional scores, including the CF score evaluated by the EORTC QLQ-C30 questionnaires from 13 patients before treatment and 12 months after treatment. The flattened boxplot 12 months after treatment shows that all 13 patients reported the highest score of CF which can be also described by the mean value of CF score 12 months after treatment  CF_mean= 100 ± 0.0. Therefore, the analysis provides enough evidence to show that CF score was significantly improved in patients treated with curative RCT and deep regional HT before and twelve months after treatment.

In the Figure S2 , "NS" is different from "ns" because NS refers to a p-value equal to 1 and "ns" refers that p-value is smaller than 1 but higher than 0.05. The p-values for SF were corrected according to Figure S1

  • Consider removing Figures S3 and S4.

We agreed with the reviewer that these figures might be too much information and thus we removed Figures S3 and S4.

  • Line 233: Again 30 patients for palliative (31?).

We added the following, line 264 in the revised manuscript:

", who did return the questionnaire after three months.. "

  • Line 242: The significance for FI comes from the comparison of only two point in the group after treatment. Is that really significant? Does that provide sufficient evidence?

As explained above, in the Section "2.4 Statistical analysis" is described that the change score between baseline and at a different time point was assessed using the same group of patients. In Figure 4 (Figure 3 revised manuscript), the symptoms scale scores from 32 patients treated with curative RCT and HT are shown before treatment and 12 months after treatment.

In Figure 3, the boxplots represent the score of all symptoms scale items, including the FI score in 32 patients before treatment and 12 months after treatment. The median FI score twelve months after treatment in 31 patients did not vary except for one patient whose score is indicated as an outlier (the dot) in the boxplot shown in Figure 3. Therefore, the analysis provide enough evidence to show that FI score was significantly improved in patients treated with curative RCT and HT before and twelve months after treatment.

  • Line 245: Same question as above. The significance for FI comes from the comparison of only two point in the group after treatment. Is that really significant? Does that provide sufficient evidence?

The significance of FI does not come from the comparison of only two points in the group after treatment. As explained in Section "2.4 Statistical Analysis", the answers of 30 patients were evaluated using the returned questionnaires before treatment and three months after treatment as shown in Figure 5. The data of 30 patients were used. All of them answered the questionnaire before and after treatment.

  • Lines 251-256: Consider removing Figures S5, S7 and S8. In Figure S6, be consistent with ns (or NS). Significance in FI and SL scales also come from 2 or 3 values comparison in the after twelve moths group.

We removed the Figures S5, S7 and S8 which did not show any significant results.

In the Figure S6 (updated supplementary material Figurer S3), "NS" is different from "ns" because NS refers to a p-value equal to 1 and "ns" refers that p-value is smaller than 1 but higher than 0.05. We confirm as previously that the analysis in 13 patients treated with curative  RCT and deep regional HT  was the same group of patients used to evaluate the baseline score (before treatment) vs 12 months after treatment.

  • Line 291: 51 patients treated with curative intent (total of 45 reported in the beginning of the results section-line158). Please verify and explain if different. Please explain why after three months the total number of patients is different.

We thank the reviewer for the comment to provide an explanation about the different number of patients in section "3.3 Acute toxicity analysis" in patients treated with curative intent. According to Table 1, in total not 45 patients but 52 patients were treated. The medical records of 51 out of 52 were available for the acute analysis during the treatment. The medical reports prepared for the follow up three months after treatment did contain information about the toxicity in only 45 out of 52 patients. Therefore, we added the following information in the section "3.3 Acute toxicity analysis", line 386 in the revised manuscript:

"The available medical records of 51 out of 52 patients treated with curative RCT in combination with HT …"

and additionally we added the following, line 391-393 in the revised manuscript:

"At three months after treatment, toxicity information for only 45 out of 52 patients treated with curative intent were available from the medical records. These reports showed that this group of patients… "

  • Line 299: Did you monitor any skin burns or subcutaneous burns, fat necrosis?

The skin burns or subcutaneous burns, fat necrosis including other serious harm to the patients during hyperthermia sessions are all monitored and documented as serious adverse events (SAEs). None of the patients included in this analysis had a documents SAEs. This can be explained by the safe temperature and thermal dose values achieved during hyperthermia sessions in our center as shown in Table 1.

  • Line 301: 42 patients treated with curative intent (total of 52 reported in the beginning of the results section-line158). Please verify and explain if different. Please explain why after three months the total number of patients is different.

The total number of patients treated with palliative intent is total 45 ( shown in Table 1), the documentation of acute toxicity symptoms during  the treatment were available in 42 out of 45 patients treated with palliative intent and 29 out of 45 patients after three months. As suggested from the reviewer we added a short explanation in section "3.3 Acute toxicity analysis", line 404-405 in the revised manuscript:

The acute toxicities were documented during the treatment and three months thereafter, for 42 and 29 palliative patients, respectively (Table 7).

  • Line 313/Figure 6: Please verify number of total patients and compare to line 158. What is the objective of the plot, if the overall survival of the two groups is not aimed to be compared?

The statistical analysis for the survival outcomes aims to only graphically show when the events of death occurred for both treatment which complement and provide further information to the reader than indicating only the survival rate. Furthermore, the comparison in this case is not reliable since the patients' groups are treated with curative vs palliative intent, which it makes impossible to match the patients groups for comparison.

  • Line 324/Figure 7: Please verify number of total patients and compare to line 158. What is the objective of the plot, if the local progression free survival of the two groups is not aimed to be compared?

We thank the reviewer for the suggestion. As it was written in the previous answer, the statistical analysis for the local progression free outcomes aims to only graphically show when the events of death occurred for both treatment which complement and provide further information to the reader than indicating only the local progression free survival rate. Furthermore, the comparison in this case is not reliable since the patients' groups are treated with curative vs palliative intent which it makes impossible to match the patients groups for comparison.

  • Figures S9 and S10: Could include a comparison between deep and superficial for the same group and present if significand different in the plot. Otherwise, consider remove, since it does not provide more information than the values presented in the text.

We agree with the reviewer and we removed both Figures S9 and S10.

  1. Discussion
  • Line 336/340/401: Please explain what is real world evidence. quality of life analysis is not always a real world measure? Are there studies on theoretical quality of life?

We thank the reviewer for comment. As mentioned before, we added an explanation for the term "real world" in the introduction part, line 74-78 in the revised manuscript:

"The term "real-world conditions" refers to a patient cohort where all patients treated with hyperthermia in our center were prospectively included and data were routinely collected from a variety of sources. The benefits and limitations of real world analyses in comparision to randomized clinical trials have been summarized elsewhere[27]."

  • Line 396: The number of patients was mentioned to be the innovative part of the presented study. Please be consistent and try to better explain.

We totally agree with the reviewer's comment to explain better about the limitation of the analysis. We added the following detailed explanation about the main limitations of this study in Discussion, line 528-533 in the revised manuscript:

"The main limitations of this study are the retrospective data analysis of patients with a high variability of tumor sites. In addition, not all patients did return the QoL questionnaires during follow up leading to a variability of the number of patients that could be included in the analysis at different time points after treatment. At 12 months after treatment there were 72 patients alive and 49 out of 72 returned the QoL. This might introduce an upward bias in the analysis of patients treated with curative intent. "

  • Please comment the statistical significance of tests that are performed with groups with only two or three patients and how that influences the results and conclusions of the current study. Consider also adding some information on how this type of analysis could be performed with more patient compliance.

We thank the reviewer for the comment. As we have explained above, there were no analysis performed with groups of two or three patients.

Reviewer 3 Report

Minor points:

Abstract

Line 24: This report analyses the quality….

Line 26: 3 and 12 months or three and twelve

Line 27: who were treated for both palliative and curative intent, respectively

Line 189, remove extra space

Results

In the results section 3.2.2 EF and SF first time in results section in full, otherwise people not familiar have to look back and search for what it meant (line 199).

Same for CF (line 209).

PF2, line 229

FI and SL AP, CO, DI, DY, FA, NV, PA, lines 241-243

Same for the figures, the text is written quite tiny, so than a preferences for full names instead of abbreviations. A the minimum the abbreviations should be mentioned in the figure legends. Figures should be interpretable without reading the text.

3.4 Outcomes part

Can you also indicate in dotted lines the OS for the EORTC reference cohort?

Statistics part

Since your comparing so many variables, I think its suitable to use a correction for multiple testing to define the importance of the changing variables, using bonferoni or a permutation test such as Benjamini and Hochberg.

Author Response

  1. Abstract

Line 24: This report analyses the quality….

Line 26: 3 and 12 months or three and twelve

Line 27: who were treated for both palliative and curative intent, respectively

Line 189, remove extra space

 We thank the reviewer for the recommended changes in abstract. We did incorporate all the suggested changes except of the changes in line 26 and 27. We used the conventions in scientific writing that numbers smaller than 10 are written out as whole numbers and all other numbers are written as numerals. Adding the word "both" in the sentence line 27 in abstract was reviewed by a native English researcher and also a coauthor of this analysis.

  1. Results

In the results section 3.2.2 EF and SF first time in results section in full, otherwise people not familiar have to look back and search for what it meant (line 199).

Same for CF (line 209).

PF2, line 229

FI and SL AP, CO, DI, DY, FA, NV, PA, lines 241-243

Same for the figures, the text is written quite tiny, so than a preferences for full names instead of abbreviations. A the minimum the abbreviations should be mentioned in the figure legends. Figures should be interpretable without reading the text.

We thank the reviewer for the suggestions to avoid the abbreviations in the text such that the reader can easily understand which of the EORTC QLQ-C30 scale items are mentioned. As recommended from the Reviewer #2, all scale items of EORTC QLQ-C30 questionnaire including their abbreviations were graphically presented in a Figure (please see Figure 1 in the revised manuscript), which will help the reader.

  1. 4 Outcomes part

Can you also indicate in dotted lines the OS for the EORTC reference cohort?

Unfortunately, the EORTC group provides only the reference data (Scott et al. 2018) about the distribution of QoL scores for given cancer populations with certain predefined characteristics, in particular stage and cancer site but no information about OS of the reference group.

  1. Statistics part

Since your comparing so many variables, I think its suitable to use a correction for multiple testing to define the importance of the changing variables, using bonferoni or a permutation test such as Benjamini and Hochberg.

We thank the reviewer for the suggestion to consider the issue of multiplicity of outcomes. But according to EORTC guideline entitled "Guidelines for assessing Quality of Life in EORTC clinical trials" (1), specifically Chapter 11, we performed a simple comparison using t-test for comparing e.g. global health status at two different timepoints which are also supported from. The idea of this approach is to summarize the repeated data (per patient) into one outcome. This one outcome can then be analysed using standard statistical techniques (e.g. t-test). Common summary measures are analysis at each time point separately and comparison to assess the change of QoL with dependence on time.

Additionally, the EORTC guideline also provides the solutions for multiple significance tests. We chose the first and foremost solution, our analysis identified one or two QoL outcomes as being of principal interest which were Global Health Status for patients treated with curative and palliative treatment. The analysis for both groups were performed separately. The Global Health Status was the primary outcome which is considered as the main focus of the analysis (presented also in the first section of results, see section 3.2.1 Global Health Status), and therefore there will be no problems of multiple testing. All other analyses may then be regarded as primarily hypothesis generating, and will be regarded more critically. To avoid any confusion, we added the primary outcome and secondary outcomes in the section 2.4 "Statistical analysis", line 188-189 in the revised manuscript:

"The primary outcome of the QoL analysis was Global Health Status score and the secondary outcomes were functional and symptom scale items."

  • https://www.eortc.org/app/uploads/sites/2/2018/02/clinical_trials__guidelines_qol.pdf

Reviewer 4 Report

The authors present a retrospective, single center analysis of the Quality of Life following deep or superficial hyperthermia treatments either curative or palliative. Strong points are the focus on QoL, but also reporting on oncological outcome and the relatively large number of patients, also considering the single centre analysis. Also, there is a high respons rate of the QoL forms in this analysis. Some weaker points are the high variability in tumor sites and in radiotherapy schemes. 

Minor comments:

The author report on the interval between HT and RT, have they performed an analysis on the effect of the interval  on local control, especially in the curative group?

If possible, a matched comparison group with radiotherapy only patients would be very welcome / add great value to this analysis 

Author Response

  1. General comments:

"The authors present a retrospective, single center analysis of the Quality of Life following deep or superficial hyperthermia treatments either curative or palliative. Strong points are the focus on QoL, but also reporting on oncological outcome and the relatively large number of patients, also considering the single centre analysis. Also, there is a high respons rate of the QoL forms in this analysis. Some weaker points are the high variability in tumor sites and in radiotherapy schemes. "

  1. Minor comments:

The author report on the interval between HT and RT, have they performed an analysis on the effect of the interval on local control, especially in the curative group?

The reviewer's suggestion to perform the analysis on the effect of time interval between HT and RT on local control in patients treated with curative intent is very much appreciated. Unfortunately due  to the heterogeneity of the patient cohort analyzed including the high variability in tumor sites, radiotherapy schemes and treatment intent (palliative versus curative) the analysis on time interval was not performed to avoid any biased results. Experimental studies (Overgaard, J. (1982): 325-32; Arcangeli, G et al. (1984): 4857s-4863s.) which showed that the radiosensitization effect induced by hyperthermia is dependent on radiotherapy dose and type of cancer, support that analysis of time interval would not be feasible in our heterogenous cohort. As part of the EU Horizon2020 Hyperboost project, our group designed a multinational data collection to analyze the effect of time interval on clinical outcomes in a homogenous cohort of rectal cancer patients treated with hyperthermia. 

If possible, a matched comparison group with radiotherapy only patients would be very welcome / add great value to this analysis

We agree that a matched pair analysis with a group of patients treated with radiotherapy only would be very important to analyze the effect of hyperthermia on quality of life. Because our patient cohort is very heterogenous, this matched pair analysis unfortunately will not be feasible and not be reliable due to the large difference of the covariates Though, this analysis is of high interest for  future clinical studies in the field of hyperthermia.

Additional changes (not requested by the reviewers):

  • Affiliations: correction of a middle name of one coauthor
  • Line 73: the EORTC abbreviations was introduced
  • Line 152-156: the deletion of EORTC QLQ-C30's scale items
  • Line 259: the word "items" was added
  • Line 325: an extra parenthesis was removed
  • Table 6: the small capital letter was substituted with the large capital letter
  • Table 7: the separation line between two first rows in Table 7 were removed
  • Line 500-502: the word "substantially" and ", if temperature is controlled and kept below certain limits " were added.
  • Line 543-544: the word "short and long term effects" was added
  • Line 559-582: the deletion of Figures' legends which were removed